# Community Detection via Measure Space Embedding

**Mark Kozdoba**
The Technion, Haifa, Israel
markk@tx.technion.ac.il

**Shie Mannor**
The Technion, Haifa, Israel
shie@ee.technion.ac.il

## Abstract

We present a new algorithm for community detection. The algorithm uses random walks to embed the graph in a space of measures, after which a modification of $k$-means in that space is applied. The algorithm is therefore fast and easily parallelizable. We evaluate the algorithm on standard random graph benchmarks, including some overlapping community benchmarks, and find its performance to be better or at least as good as previously known algorithms. We also prove a linear time (in number of edges) guarantee for the algorithm on a $p, q$-stochastic block model with where $p \geq c \cdot N^{-\frac{1}{2}+\epsilon}$ and $p - q \geq c' \sqrt{pN^{-\frac{1}{2}+\epsilon} \log N}$.

## 1 Introduction

Community detection in graphs, also known as graph clustering, is a problem where one wishes to identify subsets of the vertices of a graph such that the connectivity inside the subset is in some way denser than the connectivity of the subset with the rest of the graph. Such subsets are referred to as communities, and it often happens in applications that if two vertices belong to the same community, they have similar application-related qualities. This in turn may allow for a higher level analysis of the graph, in terms of communities instead of individual nodes. Community detection finds applications in a diversity of fields, such as social networks analysis, communication and traffic design, in biological networks, and, generally, in most fields where meaningful graphs can arise (see, for instance, [1] for a survey). In addition to direct applications to graphs, community detection can, for instance, be also applied to general Euclidean space clustering problems, by transforming the metric to a weighted graph structure (see [2] for a survey).

Community detection problems come in different flavours, depending on whether the graph in question is simple, or weighted, or/and directed. Another important distinction is whether the communities are allowed to overlap or not. In the overlapping communities case, each vertex can belong to several subsets.

A difficulty with community detection is that the notion of community is not well defined. Different algorithms may employ different formal notions of a community, and can sometimes produce different results. Nevertheless, there exist several widely adopted benchmarks – synthetic models and real-life graphs – where the ground truth communities are known, and algorithms are evaluated based on the similarity of the produced output to the ground truth, and based on the amount of required computations. On the theoretical side, most of the effort is concentrated on developing algorithms with guaranteed recovery of clusters for graphs generated from variants of the Stochastic Block Model (referred to as SBM in what follows, [1]).

In this paper we present a new algorithm, DER (Diffusion Entropy Reducer, for reasons to be clarified later), for non-overlapping community detection. The algorithm is an adaptation of the k-means algorithm to a space of measures which are generated by short random walks from the nodes of the graph. The adaptation is done by introducing a certain natural cost on the space of the measures. As detailed below, we evaluate the DER on several benchmarks and find its performance to be as good or better than the best alternative method. In addition, we establish some theoretical guarantees

on its performance. While the main purpose of the theoretical analysis in this paper is to provide some insight into why DER works, our result is also one of a few results in the literature that show reconstruction in linear time.

On the empirical side, we first evaluate our algorithm on a set of random graph benchmarks known as the LFR models, [3]. In [4], 12 other algorithms were evaluated on these benchmarks, and three algorithms, described in [5], [6] and [7], were identified, that exhibited significantly better performance than the others, and similar performance among themselves. We evaluate our algorithm on random graphs with the same parameters as those used in [4] and find its performance to be as good as these three best methods. Several well known methods, including spectral clustering [8], exhaustive modularity optimization (see [4] for details), and clique percolation [9], have worse performance on the above benchmarks.

Next, while our algorithm is designed for non-overlapping communities, we introduce a simple modification that enables it to detect overlapping communities in some cases. Using this modification, we compare the performance of our algorithm to the performance of 4 overlapping community algorithms on a set of benchmarks that were considered in [10]. We find that in all cases DER performs better than all 4 algorithms. None of the algorithms evaluated in [4] and [3] has theoretical guarantees.

On the theoretical side, we show that DER reconstructs with high probability the partition of the $p, q$-stochastic block model such that, roughly, $p \geq N^{-\frac{1}{2}}$, where $N$ is the number of vertices, and $p - q \geq c\sqrt{pN^{-\frac{1}{2}+\epsilon}\log N}$ (this holds in particular when $\frac{p}{q} \geq c' > 1$) for some constant $c > 0$. We show that for this reconstruction only one iteration of the $k$-means is sufficient. In fact, three passages over the set of edges suffice. While the cost function we introduce for DER will appear at first to have purely probabilistic motivation, for the purposes of the proof we provide an alternative interpretation of this cost in terms of the graph, and the arguments show which properties of the graph are useful for the convergence of the algorithm.

Finally, although this is not the emphasis of the present paper, it is worth noting here that, as will be evident later, our algorithm can be trivially parallelalized. This seems to be a particularly nice feature since most other algorithms, including spectral clustering, are not easy to parallelalize and do not seem to have parallel implementations at present.

The rest of the paper is organized as follows: Section 2 overviews related work and discusses relations to our results. In Section 3 we provide the motivation for the definition of the algorithm, derive the cost function and establish some basic properties. Section 4 we present the results on the empirical evaluation of the algorithm and Section 5 describes the theoretical guarantees and the general proof scheme. Some proofs and additional material are provided in the supplementary material.

## 2   Literature review

Community detection in graphs has been an active research topic for the last two decades and generated a huge literature. We refer to [1] for an extensive survey. Throughout the paper, let $G = (V, E)$ be a graph, and let $P = P_1, \ldots, P_k$ be a partition of $V$. Loosely speaking, a partition $P$ is a good community structure on $G$ if for each $P_i \in P$, more edges stay within $P_i$ than leave $P_i$. This is usually quantified via some cost function that assigns larger scalars to partitions $P$ that are in some sense better separated. Perhaps the most well known cost function is the modularity, which was introduced in [11] and served as a basis of a large number of community detection algorithms ([1]). The popular spectral clustering methods, [8]; [2], can also be viewed as a (relaxed) optimization of a certain cost (see [2]).

Yet another group of algorithms is based on fitting a generative model of a graph with communities to a given graph. References [12]; [10] are two among the many examples. Perhaps the simplest generative model for non-overlapping communities is the stochastic block model, see [13],[1] which we now define: Let $P = P_1, \ldots, P_k$ be a partition of $V$ into $k$ subsets. $p, q$-SBM is a distribution over the graphs on vertex set $V$, such that all edges are independent and for $i, j \in V$, the edge $(i, j)$ exists with probability $p$ if $i, j$ belong to the same $P_s$, and it exists with probability $q$ otherwise. If $q << p$, the components $P_i$ will be well separated in this model. We denote the number of nodes by $N = |V|$ throughout the paper.

Graphs generated from SBMs can serve as a benchmark for community detection algorithms. However, such graphs lack certain desirable properties, such as power-law degree and community size distributions. Some of these issues were fixed in the benchmark models in [3]; [14], and these models are referred to as LFR models in the literature. More details on these models are given in Section 4.

We now turn to the discussion of the theoretical guarantees. Typically results in this direction provide algorithms that can reconstruct,with high probability, the ground partition of a graph drawn from a variant of a $p, q$-SBM model, with some, possibly large, number of components $k$. Recent results include the works [15] and [16]. In this paper, however, we only analytically analyse the $k = 2$ case, and such that, in addition, $|P_1| = |P_2|$.

For this case, the best known reconstruction result was obtained already in [17] and was only improved in terms of runtime since then. Namely, Bopanna's result states that if $p \geq c_1 \frac{\log N}{N}$ and $p - q \geq c_2 \frac{\log N}{N}$, then with high probability the partition is reconstructible. Similar bound can be obtained, for instance, from the approaches in [15]; [16], to name a few. The methods in this group are generally based on the spectral properties of adjacency (or related) matrices. The run time of these algorithms is non-linear in the size of the graph and it is not known how these algorithms behave on graphs not generated by the probabilistic models that they assume.

It is generally known that when the graphs are dense ($p$ of order of constant), simple linear time reconstruction algorithms exist (see [18]). The first, and to the best of our knowledge, the only previous linear time algorithm for non dense graphs was proposed in [18]. This algorithm works for $p \geq c_3(\epsilon) N^{-\frac{1}{2}+\epsilon}$, for any fixed $\epsilon > 0$. The approach of [18] was further extended in [19], to handle more general cluster sizes. These approaches approaches differ significantly from the spectrum based methods, and provide equally important theoretical insight. However, their empirical behaviour was never studied, and it is likely that even for graphs generated from the SBM, extremely high values of $N$ would be required for the algorithms to work, due to large constants in the concentration inequalities (see the concluding remarks in [19]).

## 3   Algorithm

Let $G$ be a finite undirected graph with a vertex set $V = \{1, \ldots, n\}$. Denote by $A = \{a_{ij}\}$ the symmetric adjacency matrix of $G$, where $a_{ij} \geq 0$ are edge weights, and for a vertex $i \in V$, set $d_i = \sum_j a_{ij}$ to be the degree of $i$. Let $D$ be an $n \times n$ diagonal matrix such that $D_{ii} = d_i$, and set $T = D^{-1}A$ to be the transition matrix of the random walk on $G$. Set also $p_{ij} = T_{ij}$. Finally, denote by $\pi$, $\pi(i) = \frac{d_i}{\sum_j d_j}$ the stationary measure of the random walk.

A number of community detection algorithms are based on the intuition that distinct communities should be relatively closed under the random walk (see [1]), and employ different notions of closedness. Our approach also takes this point of view.

For a fixed $L \in N$, consider the following sampling process on the graph: Choose vertex $v_0$ randomly from $\pi$, and perform $L$ steps of a random walk on $G$, starting from $v_0$. This results in a length $L + 1$ sequence of vertices, $x^1$. Repeat the process $N$ times independently, to obtain also $x^1, \ldots, x^N$.

Suppose now that we would like to model the sequences $x^s$ as a multinomial mixture model with a single component. Since each coordinate $x_t^s$ is distributed according to $\pi$, the single component of the mixture should be $\pi$ itself, when $N$ grows. Now suppose that we would like to model the same sequences with a mixture of two components. Because the sequences are sampled from a random walk rather then independently from each other, the components need no longer be $\pi$ itself, as in any mixture where some elements appear more often together then others. The mixture as above can be found using the EM algorithm, and this in principle summarizes our approach. The only additional step, as discussed above, is to replace the sampled random walks with their true distributions, which simplifies the analysis and also leads to somewhat improved empirical performance.

We now present the DER algorithm for detecting the non-overlapping communities. Its input is the number of components to detect, $k$, the length of the walks $L$, an initialization partition $P =$

---
**Algorithm 1** DER
---
1: **Input:** Graph $G$, walk length $L$,
     number of components $k$.
2: Compute the measures $w_i$.
3: Initialize $P_1, \ldots, P_k$ to be a random partition such that
     $|P_i| = |V|/k$ for all $i$.
4: **repeat**
5:   (1)  For all $s \leq k$, construct $\mu_s = \mu_{P_s}$.
6:   (2)  For all $s \leq k$, set

$$P_s = \left\{ i \in V \;\mid\; s = \operatorname*{argmax}_l D(w_i, \mu_l) \right\}.$$

7: **until** the sets $P_s$ do not change
---

$\{P_1, \ldots, P_k\}$ of $V$ into disjoint subsets. $P$ would be usually taken to be a random partition of $V$ into equally sized subsets.

For $t = 0, 1, \ldots$ and a vertex $i \in V$, denote by $w_i^t$ the $i$-th row of the matrix $T^t$. Then $w_i^t$ is the distribution of the random walk on $G$, started at $i$, after $t$ steps. Set $w_i = \frac{1}{L}(w_i^1 + \ldots + w_i^L)$, which is the distribution corresponding to the average of the empirical measures of sequences $x$ that start at $i$.

For two probability measures $\nu, \mu$ on $V$, set

$$D(\nu, \mu) = \sum_{i \in V} \nu(i) \log \mu(i).$$

Although $D$ is not a metric, will act as a distance function in our algorithm. Note that if $\nu$ was an empirical measure, then, up to a constant, $D$ would be just the log-likelihood of observing $\nu$ from independent samples of $\mu$.

For a subset $S \subset V$, set $\pi_S$ to be the restriction of the measure $\pi$ to $S$, and also set $d_S = \sum_{i \in S} d_i$ to be the full degree of $S$. Let

$$\mu_S = \frac{1}{d_S} \sum_{i \in S} d_i w_i \tag{1}$$

denote the distribution of the random walk started from $\pi_S$.

The complete DER algorithm is described in Algorithm 1.

The algorithm is essentially a k-means algorithm in a non-Euclidean space, where the points are the measures $w_i$, each occurring with multiplicity $d_i$. Step (1) is the "means" step, and (2) is the maximization step.

Let

$$C = \sum_{l=1}^{L} \sum_{i \in P_l} d_i \cdot D(w_i, \mu_l) \tag{2}$$

be the associated cost. As with the usual k-means, we have the following

**Lemma 3.1.** *Either $P$ is unchanged by steps (1) and (2) or both steps (1) and (2) strictly increase the value of $C$.*

The proof is by direct computation and is deferred to the supplementary material. Since the number of configurations $P$ is finite, it follows that DER always terminates and provides a "local maximum" of the cost $C$.

The cost $C$ can be rewritten in a somewhat more informative form. To do so, we introduce some notation first. Let $X$ be a random variable on $V$, distributed according to measure $\pi$. Let $Y$ a step of a random walk started at $X$, so that the distribution of $Y$ given $X = i$ is $w_i$. Finally, for a partition $P$, let $Z$ be the indicator variable of a partition, $Z = s$ iff $X \in P_s$. With this notation, one can write

$$C = -d_V \cdot H(Y|Z) = d_V \left( -H(Y) + H(Z) - H(Z|Y) \right), \tag{3}$$

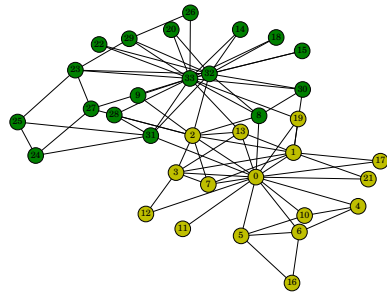
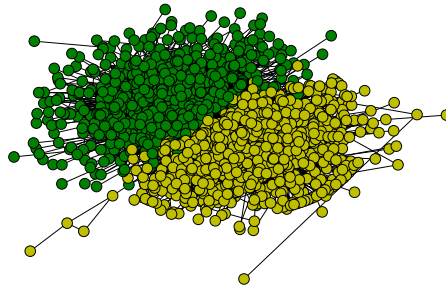

(a) Karate Club

(b) Political Blogs

where $H$ are the full and conditional Shannon entropies. Therefore, DER algorithm can be interpreted as seeking a partition that maximizes the information between current known state $(Z)$, and the next step from it $(Y)$. This interpretation gives rise to the name of the algorithm, DER, since every iteration reduces the entropy $H(Y|Z)$ of the random walk, or diffusion, with respect to the partition. The second equality in (3) has another interesting interpretation. Suppose, for simplicity, that $k = 2$, with partition $P_1, P_2$. In general, a clustering algorithm aims to minimize the cut, the number of edges between $P_1$ and $P_2$. However, minimizing the number of edges directly will lead to situations where $P_1$ is a single node, connected with one edge to the rest of the graph in $P_2$. To avoid such situation, a relative, normalized version of a cut needs to be introduced, which takes into account the sizes of $P_1, P_2$. Every clustering algorithms has a way to resolve this issue, implicitly or explicitly. For DER, this is shown in second equality of (3). $H(Z)$ is maximized when the components are of equal sizes (with respect to $\pi$), while $H(Z|Y)$ is minimized when the measures $\mu_{P_s}$ are as disjointly supported as possible.

As any $k$-means algorithm, DER's results depend somewhat on its random initialization. All $k$-means-like schemes are usually restarted several times and the solution with the best cost is chosen. In all cases which we evaluated we observed empirically that the dependence of DER on the initial parameters is rather weak. After two or three restarts it usually found a partition nearly as good as after 100 restarts. For clustering problems, however, there is another simple way to aggregate the results of multiple runs into a single partition, which slightly improves the quality of the final results. We use this technique in all our experiments and we provide the details in the Supplementary Material, Section A.

We conclude by mentioning two algorithms that use some of the concepts that we use. The Walktrap, [20], similarly to DER constructs the random walks (the measures $w_i$, possibly for $L > 1$) as part of its computation. However, Walktrap uses $w_i$'s in a completely different way. Both the optimization procedure and the cost function are different from ours. The Infomap , [5], [21], has a cost that is related to the notion of information. It aims to minimize to the information required to transmit a random walk on $G$ through a channel, the source coding is constructed using the clusters, and best clusters are those that yield the best compression. This does not seem to be directly connected to the maximum likelyhood motivated approach that we use. As with Walktrap, the optimization procedure of Infomap also completely differs from ours.

## 4   Evaluation

In this section results of the evaluation of DER algorithm are presented. In Section 4.1 we illustrate DER on two classical graphs. Sections 4.2 and 4.3 contain the evaluation on the LFR benchmarks.

### 4.1   Basic examples

When a new clustering algorithm is introduced, it is useful to get a general feel of it with some simple examples. Figure 1a shows the classical Zachary's Karate Club, [22]. This graph has a

ground partition into two subsets. The partition shown in Figure 1a is a partition obtained from a typical run of DER algorithm, with $k = 2$, and wide range of $L$'s. ($L \in [1, 10]$ were tested). As is the case with many other clustering algorithms, the shown partition differs from the ground partition in one element, node $8$ (see [1]).

Figure 1b shows the political blogs graph, [23]. The nodes are political blogs, and the graph has an (undirected) edge if one of the blogs had a link to the other. There are 1222 nodes in the graph. The ground truth partition of this graph has two components - the right wing and left wing blogs. The labeling of the ground truth was partially automatic and partially manual, and both processes could introduce some errors. The run of DER reconstructs the ground truth partition with only 57 nodes missclassifed. The NMI (see the next section, Eq. (4)) to the ground truth partition is .74.

The political blogs graphs is particularly interesting since it is an example of a graph for which fitting an SBM model to reconstruct the clusters produces results very different from the ground truth. To overcome the problem with SBM fitting on this graph, a degree sensitive version of SBM, DCBM, was introduced in [24]. That algorithm produces partition with NMI .75. Another approach to DCBM can be found in [25].

## 4.2 LFR benchmarks

The LFR benchmark model, [14], is a widely used extension of the stochastic block model, where node degrees and community sizes have power law distribution, as often observed in real graphs. An important parameter of this model is the mixing parameter $\mu \in [0, 1]$ that controls the fraction of the edges of a node that go outside the node's community (or outside all of node's communities, in the overlapping case). For small $\mu$, there will be a small number of edges going outside the communities, leading to disjoint, easily separable graphs, and the boundaries between communities will become less pronounced as $\mu$ grows.

Given a set of communities $P$ on a graph, and the ground truth set of communities $Q$, there are several ways to measure how close $P$ is to $Q$. One standard measure is the normalized mutual information (NMI), given by:

$$NMI(P, Q) = 2 \frac{I(P, Q)}{H(P) + H(Q)}, \qquad (4)$$

where $H$ is the Shannon entropy of a partition and $I$ is the mutual information (see [1] for details). NMI is equal $1$ if and only if the partitions $P$ and $Q$ coincide, and it takes values between $0$ and $1$ otherwise.

When computed with NMI, the sets inside $P, Q$ can not overlap. To deal with overlapping communities, an extension of NMI was proposed in [26]. We refer to the original paper for the definition, as the definition is somewhat lengthy. This extension, which we denote here as ENMI, was subsequently used in the literature as a measure of closeness of two sets of communities, event in the cases of disjoint communities. Note that most papers use the notation NMI while the metric that they really use is ENMI.

Figure 2a shows the results of evaluation of DER for four cases: the size of a graph was either $N = 1000$ or $N = 5000$ nodes, and the size of the communities was restricted to be either between 10 to 50 (denoted $S$ in the figures) or between 20 to 100 (denoted $B$). For each combination of these parameters, $\mu$ varied between $0.1$ and $0.8$. For each combination of graph size, community size restrictions as above and $\mu$ value, we generated 20 graphs from that model and run DER. To provide some basic intuition about these graphs, we note that the number of communities in the 1000S graphs is strongly concentrated around 40, and in 1000B, 5000S, and 5000B graphs it is around 25, 200 and 100 respectively. Each point in Figure 2a is a the average ENMI on the 20 corresponding graphs, with standard deviation as the error bar. These experiments correspond precisely to the ones performed in [4] (see Supplementary Material, Section Cfor more details). In all runs on DER we have set L = 5 and set $k$ to be the true number of communities for each graph, as was done in [4] for the methods that required it. Therefore our Figure 2a can be compared directly with Figure 2 in [4].

From this comparison we see that DER and the two of the best algorithms identified in [4], Infomap [5] and RN [6], reconstruct the partition perfectly for $\mu \leq 0.5$, for $\mu = 0.6$ DER's reconstruction scores are between Infomap's and RN's, with values for all of the algorithms above $0.95$, and for

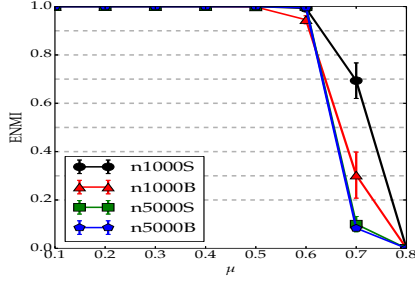 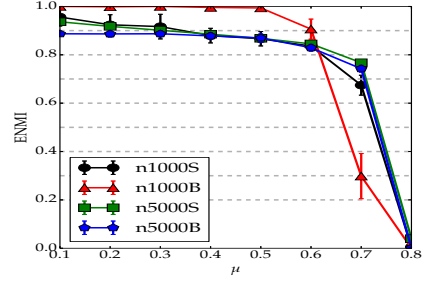

(a) DER, LFR benchmarks                    (b) Spectral Alg., LFR benchmarks

$\mu = 0.7$ DER has the best performance in two of the four cases. For $\mu = 0.8$ all algorithms have score 0.

We have also performed the same experiments with the standard version of spectral clustering, [8], because this version was not evaluated in [4]. The results are shown in Fig. 2b. Although the performance is generally good, the scores are mostly lower than those of DER, Infomap and RN.

### 4.3 Overlapping LFR benchmarks

We now describe how DER can be applied to overlapping community detection. Observe that DER internally operates on measures $\mu_{P_s}$ rather then subsets of the vertex set. Recall that $\mu_{P_s}(i)$ is the probability that a random walk started from $P_s$ will hit node $i$. We can therefore consider each $i$ to be a member of those communities from which the probability to hit it is "high enough". To define this formally, we first note that for any partition $P$, the following decomposition holds:

$$\pi = \sum_{s=1}^{k} \pi(P_s)\mu_{P_s}. \tag{5}$$

This follows from the invariance of $\pi$ under the random walk. Now, given the out put of DER - the sets $P_s$ and measures $\mu_{P_s}$ set

$$m_i(s) = \frac{\mu_{P_s}(i)\pi(P_s)}{\sum_{t=1}^{k} \mu_{P_t}(i)\pi(P_t)} = \frac{\mu_{P_s}(i)\pi(P_s)}{\pi(i)}, \tag{6}$$

where we used (5) in the second equality. Then $m_i(s)$ is the probability that the walks started at $P_s$, given that it finished in $i$. For each $i \in V$, set $s_i = \operatorname{argmax}_l m_i(l)$ to be the most likely community given $i$. Then define the overlapping communities $C_1, \ldots, C_k$ via

$$C_t = \left\{ i \in V \mid m_i(t) \geq \frac{1}{2} \cdot m_i(s_i) \right\}. \tag{7}$$

The paper [10] introduces a new algorithm for overlapping communities detection and contains also an evaluation of that algorithm as well as of several other algorithms on a set of overlapping LFR benchmarks. The overlapping communities LFR model was defined in [3]. In Table 1 we present the ENMI results of DER runs on the $N = 10000$ graphs with same parameters as in [10], and also show the values obtained on these benchmarks in [10] (Figure S4 in [10]), for four other algorithms. The DER algorithm was run with $L = 2$, and $k$ was set to the true number of communities. Each number is an average over ENMIs on 10 instances of graphs with a given set of parameters (as in [10]). The standard deviation around this average for DER was less then $0.02$ in all cases. Variances for other algorithms are provided in [10].

For $\mu \geq 0.6$ all algorithms yield ENMI of less then $0.3$. As we see in Table 1, DER performs better than all other algorithms in all the cases. We believe this indicates that DER together with equation (7) is a good choice for overlapping community detection in situations where community overlap between each two communities is sparse, as is the case in the LFR models considered above. Further discussion is provided in the Supplementary Material, Section D.

Table 1: Evaluation for Overlapping LFR. All values except DER are from [10]

| Alg. | $\mu = 0$ | $\mu = 0.2$ | $\mu = 0.4$ |
|---|---|---|---|
| DER | **0.94** | **0.9** | **0.83** |
| SVI ([10]) | 0.89 | 0.73 | 0.6 |
| POI ([27]) | 0.86 | 0.68 | 0.55 |
| INF ([21]) | 0.42 | 0.38 | 0.4 |
| COP ([28]) | 0.65 | 0.43 | 0.0 |

We conclude this section by noting that while in the non-overlapping case the models generated with $\mu = 0$ result in trivial community detection problems, because in these cases communities are simply the connected components of the graph, this is no longer true in the overlapping case. As a point of reference, the well known Clique Percolation method was also evaluated in [10], in the $\mu = 0$ case. The average ENMI for this algorithm was 0.2 (Table S3 in [10]).

## 5   Analytic bounds

In this section we restrict our attention to the case $L = 1$ of the DER algorithm. Recall that the $p, q$-SBM model was defined in Section 2. We shall consider the model with $k = 2$ and such that $|P_1| = |P_2|$. We assume that the initial partition for the DER, denoted $C_1, C_2$ in what follows, is chosen as in step 3 of DER (Algorithm 1) - a random partition of $V$ into two equal sized subsets.

In this setting we have the following:

**Theorem 5.1.** *For every $\epsilon > 0$ there exists $C > 0$ and $c > 0$ such that if*

$$p \geq C \cdot N^{-\frac{1}{2}+\epsilon} \tag{8}$$

*and*

$$p - q \geq c\sqrt{pN^{-\frac{1}{2}+\epsilon}\log N} \tag{9}$$

*then DER recovers the partition $P_1, P_2$ after one iteration, with probability $\phi(N)$ such that $\phi(N) \to 1$ when $N \to \infty$.*

Note that the probability in the conclusion of the theorem refers to a joint probability of a draw from the SBM and of an independent draw from the random initialization.

The proof of the theorem has essentially three steps. First, we observe that the random initialization $C_1, C_2$ is necessarily somewhat biased, in the sense that $C_1$ and $C_2$ never divide $P_1$ exactly into two halves. Specifically, $||C_1 \cap P_1| - |C_2 \cap P_1|| \geq N^{-\frac{1}{2}-\epsilon}$ with high probability. Assume that $C_1$ has the bigger half, $|C_1 \cap P_1| > |C_2 \cap P_1|$. In the second step, by an appropriate linearization argument we show that for a node $i \in P_1$, deciding whether $D(w_i, \mu_{C_1}) > D(w_i, \mu_{C_2})$ or vice versa amounts to counting paths of length two between $i$ and $|C_1 \cap P_1|$. In the third step we estimate the number of these length two paths in the model. The fact that $|C_1 \cap P_1| > |C_2 \cap P_1| + N^{-\frac{1}{2}-\epsilon}$ will imply more paths to $C_1 \cap P_1$ from $i \in P_1$ and we will conclude that $D(w_i, \mu_{C_1}) > D(w_i, \mu_{C_2})$ for all $i \in P_1$ and $D(w_i, \mu_{C_2}) > D(w_i, \mu_{C_1})$ for all $i \in P_2$. The full proof is provided in the supplementary material.

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
