[Supplementary Material]

# Supplementary Information to Community Detection via Measure Space Embedding

**Mark Kozdoba**
The Technion, Haifa, Israel
markk@tx.technion.ac.il

**Shie Mannor**
The Technion, Haifa, Israel
shie@ee.technion.ac.il

## A    Restarts and repeats

As any $k$-means algorithm, DER's results depend somewhat on its random initializations, and can be improved by multiple runs on the same instance with different initializations. We refer to this as restarts of the algorithm. We have observed empirically the following behaviour of DER: Suppose a graph $G$ has a ground truth partition $P_1, \ldots, P_k$. Then the output of a typical restart of DER will be a partition $C_1, \ldots, C_k$ with the property that for each $C_i, i \leq k$, either there is $j \leq k$ such that $C_i = P_j$, or there are $j_1, j_2$ such that $C_i = P_{j_1} \cup P_{j_2}$ or there are $j$ and $l$ such that $C_i \cup C_l = P_j$. In other words, DER tends to either find the precise cluster, or to glue together two original clusters, or split an original cluster into two parts. Usually most of the clusters will be found precisely, and there will be some small number of (usually small) clusters that are glued or splitted. Which clusters will be glued or splitted would depend on the random initialization. An simple way to deal with this is to use the following "repeats" strategy: Choose a number of repeats, $R$ (say, $R = 5$) and run DER $R$ times. Construct the node co-occurence matrix:

$$\hat{R}_{ij} = \text{ number of runs such that } i \text{ and } j \text{ appear in the same cluster.} \tag{1}$$

for all $i, j \in V$.

The matrix $\hat{R}$ can now be regarded as an adjacency matrix of a weighted graph and can be clustered itself. However, $\hat{R}$ will often have very clear clusters, which can be found using the following trivial threshold algorithm: Define $T = \lceil R/2 \rceil$. Initialize a set $U = V$. Choose an arbitrary $i \in U$ and define a cluster $C$ by

$$C = \{j \in U \mid \hat{R}_{ij} \geq T\}.$$

Then output cluster $C$, set $U = U \setminus C$, choose a new $i \in U$ and repeat until $U$ is empty.

While on the benchmarks a single run of DER with a single restart usually has quite high precision, repeats are a more effective way to deal with glueing and splitting than the restarts. It is of course also possible to use more sophisticated but slower algorithms instead of the threshold one to cluster the co-occurence matrix $R$.

## B    Proofs

### B.1    Lemma 3.1

*Proof Of Lemma 3.1:* The claim is obvious for step (2) of the algorithm. For step (1) the claim is implied by the following standard fact: Let $\nu_1, \nu_2, \ldots, \nu_z$ be any finite collection of measures. Set $\tilde{\nu} = \frac{1}{z} \sum_i \nu_i$. Then for any measure $\kappa$,

$$\sum_{i=1}^{z} D(\nu_i, \kappa) \leq \sum_{i=1}^{z} D(\nu_i, \tilde{\nu}). \tag{2}$$

Indeed, by rearranging terms in (2), we get

$$\sum_{j \in V} \left( \sum_{i=1}^{z} \nu_i(j) \right) (\log \tilde{\nu}(j) - \log \kappa(j)) =$$

$$z \cdot \sum_{j \in V} \tilde{\nu}(j) \left( \log \frac{\tilde{\nu}(j)}{\kappa(j)} \right) \geq 0$$

which is the non-negativity of the Kullback-Leibler divergence [1], with equality iff $\kappa = \tilde{\nu}$. □

## B.2 Main result

We now prove Theorem 5.1, which we restate here for convenience.

**Theorem B.1.** *For every $\epsilon > 0$ there exists $C > 0$ and $c > 0$ such that if*

$$p \geq C \cdot N^{-\frac{1}{2}+\epsilon} \tag{3}$$

*and*

$$p - q \geq c\sqrt{pN^{-\frac{1}{2}+\epsilon} \log N} \tag{4}$$

*then DER recovers the partition $P_1, P_2$ after one iteration, with probability $\phi(N)$ such that $\phi(N) \rightarrow 1$ when $N \rightarrow \infty$.*

Recall that a general plan of the proof was discussed in Section 5.

We note that the use of paths of length two is essential for the argument to work. Similar argument with paths of length one (edges) will not work (unless $p$ is of the order of a constant). However, we also note that paths of length two are never explicitly computed, as this would require squaring the adjacency matrix. Instead, this is achieved by considering paths of length one from the target set $C_1$ (via $\mu_{C_1}$) and paths of length one from the nodes (via $w_i$).

We proceed to implement that plan. We start with stating some preliminaries. First, we state a version of Chernoff's bound for binomial variables.

**Theorem B.2** (Theorem 2.1 in [2]). *Let $X \sim Bin(n,p)$ be a binomial variable and set $\lambda = np$. Then for all $t \geq 0$,*

$$\mathbb{P}\left(X \geq \mathbb{E}X + t\right) \leq exp\left(-\frac{t^2}{2(\lambda + t/3)}\right) \tag{5}$$

$$\mathbb{P}\left(X \leq \mathbb{E}X - t\right) \leq exp\left(-\frac{t^2}{2\lambda}\right) \tag{6}$$

In general given a binomial $X \sim Bin(n,p)$ we will often refer to $\lambda = np$ as $X$'s lambda.

The following Corollary will be useful.

**Corollary B.3** (Corrolary 2.3 in [2]). *Let $X \sim Bin(n,p)$ be a binomial variable. Then for all $\epsilon \leq \frac{3}{2}$,*

$$\mathbb{P}\left(|X - \mathbb{E}X| \geq \epsilon \cdot \mathbb{E}X\right) \leq 2exp\left(-\frac{\epsilon^2}{3}\mathbb{E}X\right) \tag{7}$$

We will also often use the following Corollary of Theorem B.2.

**Corollary B.4.** *There is a constant $c > 0$ such that the following holds: Let $X \sim Bin(n,p)$ be a binomial variable such that $\lambda = np > 1$. Then for any $N > 0$,*

$$\mathbb{P}\left(|X - \mathbb{E}X| \geq 20 \cdot \sqrt{\lambda} \cdot \log N\right) \leq c/N^2. \tag{8}$$

We now present a series of Lemmas about random graphs in the $p, q$- SBM model and about random initializations. Throughout $G = (V, E)$ will be assumed to be a random graph from the $p, q$-SBM and we denote this $G \sim \mathcal{G}_{p,q}$. Recall that $N = |V|$ is the size of the node set, and for a node $i \in V$ in a fixed graph $G$, $n_i$ is the set of neighbours of $i$, and $d_i = |n_i|$ is the degree of $i$. Also, for a set $S \subset V$, its full degree is $d_S = \sum_{i \in S} d_i$. Next, for a set $S \subset V$, we denote by $d(i, S) = |n_i \cap S|$ the number of edges between $i$ and $S$ and for two sets, $S, T \subset V$ define $d(S, T) = \sum_{i \in S} d(i, T)$ to be the number of edges between $S$ and $T$. Finally, set $d_2(i, T) = d(n_i, T)$ to be the number of paths of length two that start at $i$ and end at $T$.

In addition, let $C_1, C_2$, with $|C_1| = |C_2| = N/2$, be a random partition of $V$ into two sets, the initialization of DER. Denote $N_1 = |C_1 \cap P_1|$, and $N_2 = N/2 - N_1 = |C_1 \cap P_2| = |C_2 \cap P_1|$. We assume without loss of generality that $N_1 \geq N_2$, and set $\Delta N = N_1 - N_2$. The partition $C_1, C_2$ will be considered fixed in all the lemmas that concern the random graphs.

We proceed to give bounds on the expectations and concentration intervals of several quantities related to our problem.

For a fixed node $i \in V$, the degree $d_i$ is distributed as a sum of two independent binomials,

$$d_i \sim Bin(N/2 - 1, p) + Bin(N/2, q), \tag{9}$$

the first term counts the edges to the component to which $i$ belongs, the second to the other component. In particular, the expected degree is

$$\mathbb{E}d_i = (N/2 - 1)p + (N/2)q. \tag{10}$$

**Lemma B.5** (Degree bounds). *Let $G \sim \mathcal{G}_{p,q}$. There exists a constant $\hat{c}_1$ such that the following holds: Assume that*

$$Np \geq 100 \log N. \tag{11}$$

*Then with probability at least $1 - \hat{c}_1/N$, for all $i \in V$*

$$\frac{1}{4} \cdot \frac{N}{2} p \leq d_i \leq 2 \cdot Np. \tag{12}$$

*Proof.* Fixed a node $i \in V$, and let $X \sim Bin(N/2 - 1, p)$ and $Y \sim Bin(N/2, q)$ be two independent binomials such that $d_i \sim X + Y$. By applying (7) to $X$ with $\epsilon = \frac{1}{2}$, we obtain that

$$\frac{1}{4}\frac{N}{2}p \leq \mathbb{E}X - \frac{1}{2}\mathbb{E}X \leq X < d_i \tag{13}$$

with probability at least $1 - 2exp(-\frac{1}{12}(\frac{N}{2}p - 1))$. Using the assumption (11), it follows that there is $c > 0$ such that $2exp(-\frac{1}{12}(\frac{N}{2}p - 1)) \leq c/N^2$. Using the union bound we therefore conclude that

$$\frac{1}{4}\frac{N}{2}p \leq d_i \tag{14}$$

holds for all nodes $i \in V$ with probability at least $1 - c/N$. Similarly, we use (7) to obtain that $X \leq Np$ with probability at least $1 - c/N^2$, perhaps with a different $c$ and that $Y \leq Np$ with probability at least $1 - c'/N^2$, because $q < p$. By the union bound it follows that $d_i = X + Y \leq 2Np$ with probability at least $1 - (c + c')/N^2$, and by the union bound again, we obtain $d_i \leq 2Np$ for all $i \in V$, with probability ate least $1 - c''/N$. $\square$

In what follows we will often encounter situations where we need to bound fluctuations of sums of a fixed number of not necessarily independent random variables, and considerations similar to those in Lemma B.5 will often be omitted.

We now consider the degree of $C_1$, $d_{C_1}$. Note that by symmetry $\mathbb{E}d_{C_1} = \mathbb{E}d_{C_2}$, and that the total degree of the graph satisfies $d_G = d_{C_1} + d_{C_2}$. Therefore

$$\mathbb{E}d_{C_1} = \frac{1}{2}d_G = N\mathbb{E}d_i = N\left((N/2 - 1)p + (N/2)q\right). \tag{15}$$

The next lemma concerns the concentration of the degree of $C_1$.

**Lemma B.6.** *Set $\lambda = N^2 p$. There exist constants $\hat{c}_3, \hat{c}_4$ such that with probability at least $1 - \hat{c}_3/N$,*

$$|d_{C_1} - \mathbb{E}d_{C_1}| \leq \hat{c}_4 \log N \cdot \sqrt{\lambda}. \tag{16}$$

*Proof.* For $l, s \in \{1, 2\}$, set $S_{ls} = C_l \cap P_s$. Observe that $d_{C_1}$ can be written as

$$\begin{aligned}
d_{C_1} = 2 \cdot d(S_{11}, S_{11}) + 2 \cdot d(S_{12}, S_{12}) + 2 \cdot d(S_{11}, S_{12}) + \\
+ d(S_{11}, S_{21}) + d(S_{11}, S_{22}) + \\
+ d(S_{12}, S_{21}) + d(S_{12}, S_{22}).
\end{aligned}$$

Note that each of the terms in the sum above is a binomial variable with lambda that is smaller or equal to $cN^2 p$ for some constant $c > 0$. Therefore by applying Corollary B.4 to each term and using union bound, we obtain the result. □

The next Lemma provides an upper bound on $\Delta N$.

**Lemma B.7.** *There are constants $c_1, c_2 > 0$ such that*

$$\Delta N \leq c_1 \sqrt{N} \log N \tag{17}$$

*with probability at least $1 - c_2/N$.*

*Proof.* For the purposes of this lemma we do not assume that $N_1 > N_2$. Recall that $N_1$ is the size of the intersection $P_1$ with a random subset of $V$ of size $N/2$, denoted $C_1$. Hence $N_1$ has has the hypergeometric distribution. Set

$$\lambda = \mathbb{E}N_1 = \frac{|P_1||C_1|}{|V|} = \frac{1}{4}N. \tag{18}$$

The hypergeometric distribution satisfies concentration inequalities similar to those satisfied by the binomials. Specifically, by Theorem 2.10 in [2], the conclusion of Corollary B.4, inequality (8) holds for hypergeometric variables, with $\lambda$ is defined as in (18). The result follows by an application of that inequality. □

We now examine the quantity $d(j, C_2)$ for a node $j \in V$. The expectations satisfy

$$\mathbb{E}d(j, C_2) = N_2 p + N_1 q \quad \text{if } j \in P_1 \tag{19}$$
$$\mathbb{E}d(j, C_2) = N_1 p + N_2 q \quad \text{if } j \in P_2. \tag{20}$$

This follows from the decomposition of $d(j, C_2)$ as a sum of two binomials. Similar expressions hold also for $d(j, C_1)$. Note that when, for instance $j \in P_1$, in fact $\mathbb{E}d(j, C_2) = N_2 p + N_1 q$ if $j \in C_1 \cap P_1$, and $\mathbb{E}d(j, C_2) = (N_2 - 1)p + N_1 q$ if $j \in C_1 \cap P_1$. Since we will be interested only in orders of magnitude, we will disregard the difference between the two cases in what follows. Throughout the proof we denote

$$L = N_2 p + N_1 q \tag{21}$$

as a convenient shorthand for $\mathbb{E}d(j, C_2)$ (when $j \in P_1$).

The quantities in the following Lemma will be relevant in what follows:

**Lemma B.8.** *Assume that the partition $C_1, C_2$ is such that*

$$\Delta N \leq c\sqrt{N} \log N. \tag{22}$$

*Then there exist constants $c_1, c_2, c_3, c_4 > 0$ and $\kappa_1 > 0$ such that if $Np > \kappa_1$ then with probability at least $1 - \frac{c_1}{N}$ the following holds: For all $j \in V$,*

$$d(j, C_2) \geq c_2 Np \tag{23}$$
$$|d(j, C_1) - d(j, C_2)| \leq c_3 \sqrt{Np} \log N \tag{24}$$
$$d(j, C_1)/d(j, C_2) \geq \frac{1}{2} \tag{25}$$
$$|d(j, C_2) - L| \leq c_4 \sqrt{Np} \log N. \tag{26}$$

*Proof.* We show that the statements hold for every $j \in V$ individually with probability at least $1 - c_4/N^2$, from which the claim of the Lemma follows by the union bound.

Using inequality (8), we obtain that with probability at least $1 - c_5/N^2$,

$$|d(j, C_2) - \mathbb{E}d(j, C_2)| \leq c_6 \sqrt{Np} \log N, \tag{27}$$

and similarly

$$|d(j, C_1) - \mathbb{E}d(j, C_1)| \leq c_6 \sqrt{Np} \log N, \tag{28}$$

where in a way similar to the proof of Lemma B.5, we have used the decomposition of $d(j, C_l)$ into two binomials and the fact that $q < p$.

Assume that $Np$ is large enough so that

$$c_6 \sqrt{Np} \log N \leq \frac{1}{10} Np \tag{29}$$

holds.

By using the assumption (22) and (19) or (20), we obtain that

$$\mathbb{E}d(j, C_2) \geq \frac{1}{4} Np$$

for all $N \geq \kappa_2$ for some constant $\kappa_2 > 0$. Combining this with (27) and with (29), we obtain

$$d(j, C_2) \geq \mathbb{E}d(j, C_2) - c_6 \sqrt{Np} \log N \geq \left(\frac{1}{4} - \frac{1}{10}\right)Np, \tag{30}$$

thereby proving (23). Next, using (19), (20) and similar expressions for $d(j, C_1)$ we obtain that

$$|\mathbb{E}d(j, C_1) - \mathbb{E}d(j, C_2)| = \Delta N(p - q). \tag{31}$$

Using (31) with (27) and (28), it follows that

$$|d(j, C_1) - d(j, C_2)| \leq c\Delta Np + c' \sqrt{Np} \log N \leq c_8 \sqrt{Np} \log N, \tag{32}$$

for appropriate constants $c, c' > 0$. This proves (24. Similarly, the claim (26) holds for all $j \in P_1$ and for $j \in P_2$ we have

$$\begin{aligned} |d(j, C_2) - L| &\leq & |L - \mathbb{E}d(j, C_2)| + c'' \sqrt{Np} \log N \leq \\ &\leq & c\Delta Np + c'' \sqrt{Np} \log N \leq c_9 \sqrt{Np} \log N. \end{aligned}$$

Thus (26) holds for all $j \in V$. Finally, to show (25) write

$$\frac{d(j, C_1)}{d(j, C_2)} = 1 - \frac{d(j, C_1) - d(j, C_2)}{d(j, C_2)}. \tag{33}$$

Then (25) holds if $\left|\frac{d(j,C_1)-d(j,C_2)}{d(j,C_2)}\right| \leq \frac{1}{2}$ holds, which in turn holds by (23) and (24) , for $N$ and $Np$ larger than some fixed constant. $\square$

We now provide some estimates on the number of length two paths (which we also referr to as 2-paths in what follows).

**Lemma B.9.** *For a node $j \in P_1$,*

$$\mathbb{E}d_2(j, C_1) = \frac{1}{2}N\left(N_1 p^2 + 2pqN_2 + N_1 q^2\right) \tag{34}$$

$$\mathbb{E}d_2(j, C_2) = \frac{1}{2}N\left(N_2 p^2 + 2pqN_1 + N_2 q^2\right) \tag{35}$$

*Proof.* For $l, s \in \{1, 2\}$, set $S_{ls} = C_l \cap P_s$. There are four types of 2-paths from $j$ to $C_1$. Those that land in $P_1$ at first step, and then land at $S_{11}$. We denote paths of this type by $P_1 S_{11}$. There exist $\frac{1}{2}N \cdot N_1$ such possible paths and each one exists in $\mathcal{G}_{p,q}$ model with probability $p^2$. For some concrete path of type $P_1 S_{11}$, say $p = j, u, v$, with $u \in P_1$ and $v \in S_{11}$, let $E_p$ be the event that this path exists in the graph. The number of such paths is then $\sum_{p \in P_1 S_{11}} \mathbf{1}_{E_p}$ and the expected number of such paths is therefore $\frac{1}{2}NN_1 p^2$. The other path types are $P_1 S_{12}$, $P_2 S_{11}$, $P_2 S_{12}$, with expected numbers of paths $\frac{1}{2}NN_2 pq, \frac{1}{2}NN_1 q^2$ and $\frac{1}{2}NN_2 pq$ respectively. Hence (34) holds. Similar considerations yield (35). $\square$

Next we obtain concentration bounds on $d_2$.

**Lemma B.10.** *There are constants $c_1, c_2 > 0$, such that with probability at least $1 - c_1/N$ the following holds: For all $i \in P_1$,*

$$|d_2(i, C_1) - \mathbb{E}d_2(i, C_1)| \leq c_1 Np \log N \qquad (36)$$
$$|d_2(i, C_2) - \mathbb{E}d_2(i, C_2)| \leq c_1 Np \log N \qquad (37)$$

*Proof.* Let $n_i$ be the neighbourhood of $i$ in $G$. Set as before $S_{ls} = C_l \cap P_s$ for $l, s \in \{1, 2\}$ and set also $A_{ls} = S_{ls} \cap n_i$. Similarly to the arguments in the previous Lemmas, to obtain concentration bounds on $d_2(i, C_1)$ we represent it as a sum of binomials

$$d_2(i, C_1) = \sum_{l,s\in\{1,2\}} \sum_{t,r\in\{1,2\}} d(A_{ls}, S_{tr}).$$

Then one observes that the lambda of each such binomial is of the order $Np \cdot N \cdot p$, because the size of $A_{ls}$ is of the order of $Np$ and the size of $S_{tr}$ is of the order of $N$. Then the conclusion follows by inequality (8). Since the sets $A_{ls}$ are random sets, to carry the above argument precisely we first condition on the neighbourhood of $n_i$ and ensure (using (7)) that the sets $A_{ls}$ are indeed not larger that $cNp$ for an appropriate $c > 0$. The full details are straightforward but somewhat lengthy and are omitted. $\qquad \square$

We will also make use of the following inequalities:

$$\log(1 + t) \leq t \text{ for all } t \geq -1 \qquad (38)$$
$$t - t^2 \leq \log(1 + t) \text{ for all } t \geq -\tfrac{1}{2} \qquad (39)$$
$$|\log \frac{t}{s}| \leq \frac{|t - s|}{\min\{t, s\}} \text{ for all } t, s > 0 \qquad (40)$$
$$|\frac{s}{t + \theta} - \frac{s}{t}| = |\frac{\theta}{t + \theta}| \cdot |\frac{s}{t}| \text{ for all } t, s, \theta \qquad (41)$$

*Proof of Theorem B.1:* For $x \in V$, denote by $n_x$ the set of neighbours of $x$ in $G$. As indicated earlier, we shall use that fact that $C_1$ is slightly biased towards either $P_1$ or $P_2$. Specifically, set $\delta = \tfrac{1}{2}\epsilon$ and assume throughout the proof, without loss of generality, that $N_1 > N_2$. Then the following holds with high probability:

$$\Delta N = N_1 - N_2 \geq N^{\frac{1}{2} - \delta}. \qquad (42)$$

Indeed, note that $N_1$, as a function of the random partition, is hypergeometrically distributed with mean $N/4$ and standard deviation of order $N^{\frac{1}{2}}$. Hence, by the central limit theorem for the hypergeometric distribution (see [3]; [4]),

$$\mathbb{P}\left(\left|N_1 - \frac{1}{4} \cdot N\right| \geq N^{\frac{1}{2} - \delta}\right) \to 1 \qquad (43)$$

with $N \to \infty$. Statement (43) guarantees a deviation from the mean, and in particular that (42) holds with high probability.

To prove the Theorem we now establish the following claim:

**Claim B.11.** *Fix a partition $C_1, C_2$ of $V$, satisfying eq. (42) and (22). Under assumptions (3) and (4), with probability at least $1 - \frac{1}{N}$ graph $G$ satisfies: For all $i \in P_1$,*

$$D(w_i, \mu_{C_1}) > D(w_i, \mu_{C_2}). \qquad (44)$$

Note that the assumptions of the Claim depend only on randomness of the partitions and are satisfied with high probability. Indeed, (42) holds as discussed above and (22) follows from Lemma (B.7).

Once we prove the claim, by symmetry we will also have for all $i \in P_2$ a reverse inequality in (44), and together with (42) this will prove the theorem. We proceed to prove the claim.

Observe that by definition we have $\mu_{C_l}(i) = \frac{d(i, C_l)}{d_{C_l}}$ for every $i \in V$.

Therefore we can rewrite (44) as:

$$\sum_{j \in n_i} \log \frac{d(j, C_1)}{d(j, C_2)} + \tag{45}$$

$$+ d_i \log \frac{d_{C_2}}{d_{C_1}} \tag{46}$$

$$> 0 \tag{47}$$

We now bound the term (46). Using (40) we obtain

$$\left| \log \frac{d_{C_2}}{d_{C_1}} \right| \leq \frac{|d_{C_2} - d_{C_1}|}{\min\{d_{C_2}, d_{C_1}\}}. \tag{48}$$

Using (15) and (16) we obtain that

$$\min\{d_{C_2}, d_{C_1}\} \geq cN^2 p, \tag{49}$$

and that

$$|d_{C_2} - d_{C_1}| \leq cN \log N \sqrt{p}. \tag{50}$$

In addition, recall that by Lemma B.5, $d_i \leq cNp$. Therefore we obtain that

$$\left| d_i \log \frac{d_{C_2}}{d_{C_1}} \right| \leq cNp \frac{N \log N \sqrt{p}}{c'' N^2 p} \leq c''' \log N \sqrt{p} \leq c''' \log N \tag{51}$$

for some constant $c''' > 0$.

We now examine the term (45). Using (39), write

$$\log \frac{d(j, C_1)}{d(j, C_2)} \geq \frac{d(j, C_1) - d(j, C_2)}{d(j, C_2)} - \left( \frac{d(j, C_1) - d(j, C_2)}{d(j, C_2)} \right)^2. \tag{52}$$

Note that by (25), $\frac{d(j, C_1)}{d(j, C_2)} \geq \frac{1}{2}$ and therefore (39) applies. We now replace the denominator in the first term of the right hand of (52) by a quantity independent of $j$, namely by $L$ as defined in (21). Using (41) with $s = d(j, C_1) - d(j, C_2)$, $t = L$ and $\theta = d(j, C_2) - L$, write

$$\frac{d(j, C_1) - d(j, C_2)}{d(j, C_2)} \geq \frac{d(j, C_1) - d(j, C_2)}{L} - \frac{|d(j, C_2) - L|}{d(j, C_2)} \cdot \frac{|d(j, C_1) - d(j, C_2)|}{L}. \tag{53}$$

To summarize, we have obtained that

$$\sum_{j \in n_i} \log \frac{d(j, C_1)}{d(j, C_2)} \qquad\qquad \geq \tag{54}$$

$$\sum_{j \in n_i} \frac{d(j, C_1) - d(j, C_2)}{L} \tag{55}$$

$$- \sum_{j \in n_i} \frac{|d(j, C_2) - L|}{d(j, C_2)} \cdot \frac{|d(j, C_1) - d(j, C_2)|}{L} \tag{56}$$

$$- \sum_{j \in n_i} \left( \frac{d(j, C_1) - d(j, C_2)}{d(j, C_2)} \right)^2. \tag{57}$$

Note that the term (55) satisfies

$$\sum_{j \in n_i} \frac{d(j, C_1) - d(j, C_2)}{L} = \frac{d_2(i, C_1) - d_2(i, C_2)}{L}. \tag{58}$$

This term counts the number of 2-paths and is the heart of the proof. Before analysing it, we bound the other two terms in the inequality in (54). Plugging in the estimates from Lemma B.8, we obtain for (56) that

$$\sum_{j \in n_i} \frac{|d(j, C_2) - L|}{d(j, C_2)} \cdot \frac{|d(j, C_1) - d(j, C_2)|}{L} \leq c \cdot d_i \frac{\sqrt{Np} \log N}{Np} \cdot \frac{\sqrt{Np} \log N}{Np}. \tag{59}$$

Using the degree estimate form Lemma B.5, $d_i \leq cNp$, we thus get

$$\sum_{j \in n_i} \frac{|d(j, C_2) - L|}{d(j, C_2)} \cdot \frac{|d(j, C_1) - d(j, C_2)|}{L} \leq c(\log N)^2 \tag{60}$$

for an appropriate $c > 0$. Similarly, for the term (57) we have

$$\sum_{j \in n_i} \left( \frac{d(j, C_1) - d(j, C_2)}{d(j, C_2)} \right)^2 \leq c \cdot d_i \cdot \frac{Np \log^2 N}{N^2 p^2} \leq c \cdot \log^2 N, \tag{61}$$

with some (perhaps different) $c > 0$.

We now proceed to obtain a lower bound on (58). The crucial property of length two path counts, $d_2(i, C_1)$ and $d_2(i, C_2)$, that enables such a bound is that the difference between the expectations of these quantities is of larger order of magnitude than their fluctuations.

Indeed, by Lemma B.10, with probability at least $1 - c/N$ we have that

$$d_2(i, C_1) - d_2(i, C_2) \geq \mathbb{E}d_2(i, C_1) - \mathbb{E}d_2(i, C_2) - 2cNp \log N \tag{62}$$

for all $i \in P_1$. In addition, by Lemma B.9,

$$\mathbb{E}d_2(i, C_1) - \mathbb{E}d_2(i, C_2) = \frac{1}{2}N\Delta N(p - q)^2 \geq N^{3/2 - \delta}(p - q)^2, \tag{63}$$

where we have used (42) in the last inequality.

Incorporating the inequalities (51), (60), (61), we obtain that $D(w_i, \mu_{C_1}) > D(w_i, \mu_{C_2})$ holds if the following inequality holds:

$$\frac{N^{3/2 - \delta}(p - q)^2 - 2cNp \log N}{L} - c \log N > 0. \tag{64}$$

To prove the theorem, it remains to choose $p$ and $q$ such that (64) is satisfied. Such $p, q$ are given by the assumptions (3), (4). Indeed, recall that $L$ satisfies $L \leq cNp$ for an appropriate $c > 0$ and hence under assumptions (3), (4) we have

$$\frac{N^{3/2 - \delta}(p - q)^2 - 2cNp \log N}{L \log N} \to \infty \tag{65}$$

with $N \to \infty$, hence yielding (64). $\qquad\square$

## C  LFR benchmarks

In this section we specify the full parameters used for the experiments in the paper.

The LFR model is generated from the following parameters: The graph size $N$, the mixing parameter $\mu$, community size lower and upper bounds $c_{min}, c_{max}$, average degree $d$, maximal degree $d_{max}$, and the power law exponents for the degree and community size distributions - which are in all cases set to their default values of $-2$ and $-1$ respectively. In addition, in the overlapping case, parameter $n$ specifies the number of nodes that will participate in multiple communities, and the parameter $m$ specifies the number of communities in which each such node will participate.

The LFR models were generated using the software available at [5].

For the non overlapping LFR benchmarks we have used $d = 20$ and $d_{max} = 50$, with the rest of parameters as specified in Section 4.2. This corresponds precisely to the experiments in [6]. The repeats strategy is described in Section A. For each given graph instance, DER was run with 15 repeats, using 3 restarts in each run. The results of the repeats were clustered using the threshold algorithm described in Section A, except in the $\mu = 0.7$ in which we have used the spectral clustering to cluster the co-occurence matrix.

The LFR experiments with the spectral clustering algorithm that are shown in Figure 2.b were performed using the spectral clustering version in Python sklearn v0.14.1 package, which is an implementation of the algorithm in [7]. The spectral clustering was run with 150 restarts of its final stage Euclidean k-means step. We note that while the repeats strategy could be applied to the spectral clustering too, it did not improve the performance in this case (despite the fact that different runs of spectral clustering returned somewhat different results). The results shown in Figure 2.b are without repeats.

For the overlapping community benchmarks we have used the following settings: $N = 10000$, $d = 60$, $d_{max} = 100$, $c_{min} = 200$, $c_{max} = 500$. The value of $n$ was $5000$ and $m$ was $4$. These are the settings that were used in [8]. As discussed in the next section, in one sense these settings can be considered a heavy overlap, while there is a different sense in which they can be considered sparse. In all cases we have run DER with 15 repeats and 3 restarts per run, and we have used the spectral clustering to cluster the co-occurence matrix.

Recall that our approach to overlapping communities is to first obtain a non-overlapping clustering and then to post-process it to obtain overlapping communities. One can ask therefore what will happen if in the non- overlapping step, DER is replaced by another non-overlapping clustering algorithm. We have tried using spectral clustering instead of DER, and applied the same post-processing. In all cases this resulted in ENMI values close to 0.

## D   Overlapping LFR benchmarks

We refer to [9] and [10] for the definitions of the LFR models. In this section we make a few brief comments regarding the structure of the overlapping LFR communities.

To simplify the discussion, we restrict our attention to the particular settings that were used in the evaluation in Section 4.3. The settings $n = 5000$ and $m = 4$ (see Section C) imply that there are 5000 such that each node belongs to a single community, and 5000 nodes such that each node belongs to 4 communities. These settings may be considered as a heavy overlap (see [11]). Indeed, it follows theoretically from the way LFR communities are generated, and also is observed in actual graphs, that under these settings each community $C$ contains about $20\%$ of nodes that belong only to $C$, and each of the remaining $80\%$ of the nodes belongs to $C$ and to 3 other communities.

On the other hand, for a node $i \in C$ that belongs to 3 other communities, the 3 other communities are chosen at random among about 75 remaining communities of the graph. This implies that for each pair of communities $C, J$, the intersection between them is small and if a node $i \in V$ is chosen at random, the event $i \in C$ is almost independent of the event $i \in J$.

The above small intersections and lack of correlations between communities property implies that random walk started from community $C$, after two steps has a chance of about $1/16$ of returning to $C$ while the rest of the probability is distributed more or less uniformly between the other communities (and is much less than $1/16$ for each community that is not $C$). In other words, the measures $w_i$ and $w_j$ have much more chance of being correlated if $i$ and $j$ belong to some common $C$ than otherwise. This explains why DER works well on these graphs.