[Reviews · NeurIPS 2015]

Submitted by Assigned_Reviewer_1

. I have two concerns with this paper:

First, I think SBM is not big at NIPS and even though people are interested in community finding, I do not know if this paper would be a good fit.

Second, the algorithm seems to require a very large value of p, which is the connection rate between the elements belonging to the same class. The authors show that p scales with 1/sqrt(N) when in the literature p typically scales as log(N)/N, which is a far more stringent condition. Also there are some recent papers by Abbe and coauthors in which they prove the conditions for recovery and they show an efficient algorithm that works very close to the proposed bounds. I would have expected that the authors compare with these works (http://arxiv.org/abs/1405.3267, http://arxiv.org/abs/1503.00609 and http://arxiv.org/abs/1506.03729).

Summary: In this paper the authors propose a new algorithm for solving the stochastic block model with known guarantees and that it performs well with the competing algorithms.

Submitted by Assigned_Reviewer_2

This paper introduces a new approach to community detection which attempts to find a partition of a network into communities along with a probability distribution for each community for the set of nodes visited in an L-step random walk, such that these probabilities explain well the actual probabilities of the nodes visited when started from a node in that community.

As far as I am aware this is an original approach. It is computationally highly efficient, and performs remarkably well in practice. A caveat with the empirical results though is that the parameter L is tuned differently in each of the sets of experiments, and no guidance is given on how to tune it -- indicating that the datasets are likely to be overfitted somewhat.

A weakness is that the empirical results from other methods have not been replicated, but simply copied from these other papers. This may introduce errors (e.g. different implementation of the evaluation metric).

An extension towards overlapping clusters is provided as well, which also performs well empirically.

The paper is well-presented.
Summary: This is an interesting paper on a novel approach for community detection. The approach is simple, elegant, and well-motivated, computationally efficient, comes with a theoretical guarantee (in a rather strongly idealized setting), and appears very competitive empirically.

Submitted by Assigned_Reviewer_3

This paper presents a graph partitioning algorithm for community detection based on random walk. It represents each node in the graph by a distribution, which is induced by performing a fixed-length random walk on the graph starting from that node. The nodes are the clustered by a kmeans-style algorithm. Empirical results are presented on some benchmark graphs for detecting non-overlapping and overlapping communities.

The results show similar or better performance compared to two leading algorithms identified by a recent empirical published in 2009. A theoretical result is also presented that shows in some limited situations (k=2, identical partition size, 1-step random walk) the proposed algorithm with high probability converges to the true partition in one iteration.

The algorithm is interesting and appears to be novel in how random walk is specifically used. I particularly liked the different interpretations that were presented by the paper in the objective (eg. equation 3).

With that said, the use of random walk for graph partitioning has been considered in many studies, including walktrap (while different in details, this method is conceptually quite similar to the proposed method) and infomap as referenced by the paper. The paper discusses the distinction between the proposed approach and these two other methods, but I felt that the discussion stops a bit short in providing a clear argument as to why the proposed method should be preferred to the existing methods.

The experimental results seem competitive (for the overlapping case). But I do have a few concerns. For the non-overlapping case, L is set to be 5 and for the overlapping case L is set to 2. Why use different L's? How are they decided? How sensitive is the method to the choice of L?

A minor presentation issue. Figure 2 does not have a caption. For comparison purposes, results of the competing methods should be included in the paper other than pointing the readers to a different papers to make the comparison.

For the theoretical result, I was not able to carefully read through the supplementary material to verify the full proof. Based on the stated result (theorem 5.1), I find it to be limited in the cases that it considers. That is, the only cases the result is applicable is when

k=2, and when the two clusters are of exact equal size. The equal-size assumption seems particularly restrictive to me. It is also stated that the result is only for L=1 (random walk of one step) and it is unclear how it would generalize to longer walks, which is used in the experiments.

The presentation of the algorithm has some confusing parts to me. Algorithm 1 does not explicitly talk about random walk. I assume w_i is computed using random walk, but it is unclear to me if it is computed analytically using the stochastic matrix or by doing random walks starting from i.

D is referred to as acting as a distance function. In the traditional Kmeans algorithm, distances are to be minimized. Here, however, obj 2 is to be maximized.

Further, obj 2 is referred to as the cost. Generally, we try to minimize cost. The inconsistent use of the terminology is confusing.

Summary: The algorithm appears to be novel and competitive empirically (for their specific choice of parameter L). The theoretical result could be interesting but seems overly restrictive in its applicability.

Submitted by Assigned_Reviewer_4

I have read the paper ``community detection via measure space embedding".

The paper studies the problem of

community detection. Community detection is a well-studied topic. The paper proposes a new method called

Diffusion Entropy Reducer (DER).

The paper is interesting to some extent. I am generally positive about the paper, but I have the following concerns.

First, the theoretic analysis is for a very idealized model: the p-q stochastic block model. This is even narrower than the well-known stochastic block model, which is already relatively narrow. This is a major drawback, for there are many recent results on more realistic models, such as the Karrer and Newman's Degree Corrected Block Model (DCBM). The paper cites Karrer and Newman's model, indicating that they are aware of this model. However, they do not adequately compare with recent work on community detection for DCBM (e.g.,

Zhao, Levina, and Zhu (2012), Jin (2015)). These papers deal with the more realistic

DCMB and provide sharp convergence results, which seems to more relevant to the real world social networks.

Second, for numerical comparison, the paper also do not adequately compare with existing literature, especially those on DCBM. For example, the paper analyzes the Karate and the weblog data, but they do not

compare the error rate with many of the recent methods, including Profile Likelihood method by

Amini, Chen, Bickel and Levina (2013), Bickel and Chen (2009) and by Zhao,

Levina, and Zhu (2012),

spectral methods by Jin (2015).

Take the weblog data set for example. Several of these methods

can have an error rate lower than 58/1222, which is similar to 57/1222 presented in this paper.

Third, the paper seems to overclaim in several places. In particular,

the paper claims that ``It can also be easily checked that spectral clustering, in form given in [8], is not close to ground truth ...".

While this is true, we must note that there are many recent work on spectral clustering which gives a much better error rate. For a more fair comparison, the paper needs to compare with the most recent literature on this well-studied topic.

Fourth,

since the method is based on k-means, it is critical to compare the computation time with other methods, especially with spectral methods (which are among fastest ones).

Given two well-studied data sets, it is probably not hard to get an error rate down to somewhere around $57/1222$. Therefore, to show the new method is advantageous, it is critical to have more thorough comparison, both in methods, computing, and maybe with more real data sets.

Last, the presentation can be improved. The algorithm presented is not easy to digest.

In summary, the authors are not aware of quite a few most recent advancement on community detection for SDM and DCBM.

Therefore,

the paper not only lacks of a fair comparison with existing literature, but also tends to over claim

in several places.

Summary: The paper over-claims in several places, especially it lacks of fair comparison with most recent literature on DCBM. The model used for theoretical analysis is substantially narrower than those in some existing literature.

Author Feedback
Author rebuttal: We thank the reviewers for their attention to the paper. We would like to address in points (1),(2) below issues that were common to a number of reviews. Then points (3)-(6) concern issues related to specific
reviews.

1) As mentioned in the reviews and emphasized in the paper itself, there exist methods that apply in more general situations than Theorem 5.1. However, we would like to mention three points on this matter.

First, Theorem 5.1 has the best runtime among methods that apply where it applies. It is a *linear time* guarantee, which is *impossible* to achieve using spectral, nuclear norm, or other relaxation procedures. Yet our range of parameters is wider than what is known using other linear time methods. This is discussed on lines 121-136 in the paper.

Second, our approach simply studies a different process. The 'graph clustering guarantees' literature is dominated by so called relaxation methods. On the other hand, we study a different, more direct process. What we prove is the basic 'corner stone' case that demonstrates that analysis of the k-means-like procedures is possible in case of graph clustering. This is not a trivial statement given the scarcity of theoretical results on k-means in general. Such a result is of value to the community that is interested in theoretical guarantees. Our contributions here are the interpretation of the metric D in terms of paths, an initialization which is different from what is commonly done in k-means related work, and the analysis that exploits this initialization.

Third and foremost, our empirical results show clearly that the algorithm itself is significantly more powerful than the analytic setup considered in this paper and is competitive on LFR benchmarks - benchmarks that are used by the "practical" rather than "theoretical" communities. While as we noted above the study of our algorithm is analytically possible, these empirical results suggest that such a study can also be useful.

2) Regarding the value of L in the experiments - we used the lowest value for which the results became stable (meaning that for different random initializations, same final clusters were obtained), but this was not mentioned in the paper. Will be added.

We also note that increasing the value of L moderately above the lowest stable value would not change the results. In particular,
repeating the overlapping experiments with L=5 would result in same scores.

3) Assigned_Reviewer_2: The L>1 case in the proofs should translate to counting paths of length 2*L in the graph and to some additional nuances. This is more involved than the present 2-path case, but does not look impossible and should bring further improvement in the value of p.

4) Assigned_Reviewer_3 - We agree that comparing algorithms on only Blog and Karate graphs would not be sufficient. These graphs were intended as illustration rather than evaluation or comparison. We also made it clear that the spectral algorithm can (with k=3) extract the right components practically exactly. We did not intend to overclaim, or in fact any claim at all.

The core of our evaluation is the LFR graphs. We compared (among other methods) to the standard spectral clustering because it is the by far the most popular graph clustering algorithm and the performance it achieves is not critically bad. We thank the reviewer for the references, of which we were not aware,and we will include some of the suggested literature as remarks on the development of DCBM.

5) Review_6: We agree that it would have been safer to reproduce the results directly. However, overall, as presented our results compare against about 16 algorithms (12 non-overlapping and 4 overlapping). This is impossible to reproduce in a research paper. We did perform the spectral clustering experiments directly exactly because of this safety reason, as a sanity check. The spectral clustering results for non overlapping case are in figure 2(b). We also did an experiment with spectral clustering in the overlapping case, but the results were not competitive (details in supplementary info,lines 408-412). Also, the particular metric evaluation issue is rather safe in this case -- in all the papers the metric was evaluated using the same code (downloadable from Fortunato's site)

6) Review_7: We thank the reviewer for the references in the review. However, to the best of our understanding, none of these papers is, as of the submission time, published in a peer-reviewed venue and two of the three papers are extremely recent. We do not see how it is possible to expect us to compare our results to unpublished, unreviewed work. We will take a more detailed look at the suggested papers and add references as needed.

Please also consider our notes in point (1) regarding the generality.

Please note also that besides the theorem our results include evaluation which shows that the algorithm is competitive on quite complex non SBM benchmarks.